# Differences in experiences of care between patients diagnosed with metastatic cancer of known and unknown primaries: mixed-method findings from the 2013 cancer patient experience survey in England

Richard Wagland,[1] Mike Bracher,[2] Allison Drosdowsky,[3] Alison Richardson,[4] John Symons,[5] Linda Mileshkin,[6] Penny Schofield[7]

For numbered affiliations see end of article.

**Correspondence to**
Dr Richard Wagland;
r.wagland@soton.ac.uk

## ABSTRACT

**Objectives** To explore differences in experiences of care reported in the Cancer Patient Experience Survey (CPES) between patients with cancer of unknown primary (CUP) and those with metastatic disease of known primary (non-CUP); to determine insights pertaining to the experiences of care for CUP respondents from free-text comments.

**Design** Two separate, but related, studies, involving secondary analysis of existing data. Using frequency matching of CUP and non-CUP patients, statistical comparisons of responses to CPES questions were conducted. Free-text comments from CUP respondents were analysed thematically.

**Setting and participants** The CPES questionnaire comprises 63 closed questions measuring 8 areas that relate to experience of care and 3 free-text questions. Questionnaires were mailed to all adult patients (aged ≥16 years) in England with cancer admitted to hospital between 1 September 2013 and 30 November 2013.

**Results** Matched analysis of closed response items from 2992 patients found significant differences between CUP (n=1496) and non-CUP patients (n=1496): CUP patients were more likely to want more written information about their type of cancer and tests received, to receive their diagnosis from a general practitioner (GP) and have seen allied health professionals, but less likely to have understood explanations of their condition or had surgery. Freetext responses (n=3055) were coded into 17 categories and provided deeper insight regarding patient information and interactions with GPs. CPES data may include a preponderance of patients with favourable CUP subtypes and patients initially identified as CUP but whose primary was subsequently identified.

**Conclusions** These are the first large-scale studies to explore the experiences of care of CUP patients. The significant differences identified between the experiences of CUP and non-CUP patients suggest CUP patients require more psychosocial support and specific interventions to manage diagnostic uncertainty and the multiple investigations many CUP patients face.

### Strengths and limitations of this study

► The first large-scale study quantifying experiences of Cancer of Unknown Primary (CUP) patients compared with those with known primary cancers;

► The first qualitative analysis of free-text comments drawn from a national sample of CUP patients;

► The profile of CUP patients who responded to the Cancer Patient Experience Survey were found not to be representative of this patient population;

► Reasons for sample limitations include: time between patient identification and survey participation; inconsistent administration of International Classification of Diseases 10th Revision codes; neglect of CUP typology;

► These limitations highlight the need for prospective, observational cohort studies to investigate the experiential, informational and psychosocial issues faced by CUP patients.

Substantial limitations were identified with the CPES data, emphasising the need for prospective studies.

## INTRODUCTION

Patients with cancer of unknown primary (CUP) have metastatic malignant disease, for which an identifiable primary site has not been identified after extensive clinical evaluation. CUP is not a single disease, but rather a heterogeneous collection of cancer types with a wide variety of clinical presentations, which are hypothesised to share a common tendency to metastasise early.[1] CUP was the fifth most common cancer in the UK in 2014, accounting for 3% of new cancers and 6% of cancer deaths.[2] Worldwide it has been identified as between the sixth to eighth most common cancer, accounting for 2.3%–5%

**BMJ**

**Table 1**  Terms used in NICE guideline to define CUP

| | |
|---|---|
| MUO | Metastatic malignancy identified on the basis of a limited number of tests, without an obvious primary site, before comprehensive investigation |
| Provisional CUP origin | Metastatic epithelial or neuroendocrine malignancy identified on the basis of the histology or cytology, with no primary site detected despite a selected initial screen of investigations, before specialist review and possible further specialised investigations |
| Confirmed CUP origin | Metastatic epithelial or neuroendocrine malignancy identified on the basis of final histology, with no primary site detected despite a selected initial screen of investigations, specialist review and further specialised investigations as appropriate |

CUP, cancer of unknown primary; MUO, malignancy of undefined primary origin; NICE, National Institute of Clinical Excellence.

of all new cancer cases and is the third to fourth most common cause of death.[3] [4] The UK National Institute of Clinical Excellence (NICE) published guidelines for the management of CUP patients in 2010,[5] which for the first time developed a taxonomy of definitions that reflected different phases of investigations for CUP: malignancy of undefined primary origin (MUO); provisional CUP (pCUP) and confirmed CUP (table 1). The guidelines also recommended the establishment of specialist CUP multidisciplinary teams (MDTs) in each National Health Service (NHS) cancer centre to include an oncologist, palliative care physician and clinical nurse specialist.[5]

However, with the exception of three previous studies,[6–8] there is very little published research on quality of life (QoL) and psychosocial aspects of CUP. Patients with CUP struggle with uncertainty and distress, especially regarding prognosis,[6] possible recurrence and the primary's hereditary potential.[7] Problems with care continuity commonly faced by cancer patients were amplified for those with CUP, particularly in relation to coordination, accountability and timeliness of care.[7] A recent cross-sectional study from Greece, using matched-sample analysis, found patients with CUP experienced higher depression and anxiety and poorer QoL compared with those who have metastatic disease of either breast or colorectal cancer.[8]

These findings suggest CUP patients may have unique psychosocial and supportive care needs, which require development of targeted supportive care interventions. Internationally, routine assessments of patient experiences of care are used increasingly to drive service quality improvements.[9] [10] In the UK, the NHS Cancer Reform Strategy,[11] Outcomes Strategy for Cancer[12] and recent Cancer Taskforce[13] documents highlight the important role of patient experiences in measuring and improving clinical quality. The national Cancer Patient Experience Survey (CPES) is an extensive, UK-wide programme of research about cancer patients' experiences of care while undergoing inpatient or day-case treatment.[14] The CPES has been administered annually since 2010 and is the biggest survey programme of its kind in the world. The data are a public resource and available from the survey provider, Quality Health. This represented an opportunity to interrogate these data to understand CUP patients' experiences of care to underpin the development of targeted care interventions for this population.

Drawing from the 2013 CPES, this paper reports findings from two related studies: qualitative and quantitative. The aim of both studies was to better understand the experiences of care among CUP patients. The quantitative study described perceived experiences of care of patients diagnosed with CUP compared with patients with metastatic cancer of a comparable known primary (non-CUP). The second, qualitative study analysed free-text responses of CUP patients in the CPES to identify emerging themes and insights regarding their experiences of cancer care.

## METHODS
### Survey design
The two studies reported here consist of secondary analyses of data collected as part of the English 2013 CPES, to assess differences in responses between CUP and non-CUP patients and to explore freetext responses of patients with CUP.

The 2013 CPES was administered by the survey provider, Quality Health. The CPES questionnaire was mailed to all adult patients (aged ≥16 years) in England with a diagnosis of cancer, who had been admitted to an NHS hospital as an inpatient or day case patient over a 3-month period (1 September 2013–30 November 2013). Non-responders were sent one reminder letter and a further reminder letter with questionnaire if necessary. The overall response rate for the 2013 CPES was 64% (n=68 737).[14]

### Survey instrument
The CPES instrument contained 63 closed question items, measuring eight key areas of patient experience across the care trajectory from diagnosis to leaving hospital: access to care; respect for patients' preferences; information and education; physical comfort; emotional support; involvement of family and friends; continuity and transition and coordination of care. Three freetext comment boxes at the end of the questionnaire asked the following questions: Was there anything particularly good about your NHS care? Was there anything that could be improved? Any other comments?

### Identification of respondents
All NHS hospitals treating adult patients with cancer in England were included. Patients were identified from data provided by these organisation, selected from local patient administration systems. Patients were identified as CUP using the 10 revision of the International

Statistical Classification of Diseases and Related Problems (ICD-10) codes: C77 (secondary and unspecified malignant neoplasm of lymph nodes), C78 (secondary malignant neoplasm of respiratory and digestive organs), C79 (secondary malignant neoplasm of other and unspecified sites) and C80 (malignant neoplasm, without specification of site).[15]

## ANALYSIS

### Frequency matching analysis: matching procedure

A frequency matching analysis was conducted to compare responses to the closed survey questions between CUP and non-CUP respondents. CPES (2013) respondents of 4535 were identified as CUP patients. The original plan was to match CUP and non-CUP samples for analysis on three variables: age group in deciles, sex and type of admission (ordinary admission, day case admission or regular day case admission). After exploration of the data, two further variables needed to be accounted for: tumour type and time since treatment start. Prevalence of breast and prostate cancer were high among non-CUP respondents to the CPES questionnaire, and it seemed inappropriate to match the CUP sample against a high proportion of such patients given estimates of likely site of origin in CUP cases from autopsy and biomarker studies.[16] For this reason, the dataset was restricted so these two tumour types represented only 5% each in the known primary sample. Preliminary analysis also found a marked difference between the CUP and non-CUP respondents regarding time since commencing cancer treatment (figure 1). Patients with CUP were found disproportionally to have commenced treatment >1 year prior to the survey (34% of CUP vs 73% of non-CUP); many more than would normally be expected among a random CUP sample and suggests a high proportion of favourable CUP subtypes.[4] Due to the small proportion of non-CUP patients who responded that it had been longer than 1 year since they began treatment, it was impossible to match on this variable, so the sample was restricted to only those patients who responded 'less than 1 year'.

The final sample included 1496 CUP patients who began treatment in the past year, with no missing data on the matching variables (age, sex and administration type). All patients were assigned identification and CUP patients were matched with randomly selected non-CUP patients, who had no missing data on the matching variables, began treatment in the past year and were diagnosed with metastatic disease in specific tumour types (colorectal, breast, head and neck, kidney/adrenal, prostate, pancreas and upper and lower gastrointestinal). These sites were selected as the primary sites from which CUP is most commonly thought to arise.[16] The final combined dataset contained 2992 respondents (table 2).

### Frequency matching analysis: analysis

$\chi^2$ tests assessed associations between diagnosis type (CUP vs non-CUP) and responses to each item. Given the very large sample, the likelihood of finding statistically significant associations at $p<0.05$ was high. It was therefore determined a priori that a 'small' or greater effect would be classified as meaningful; this corresponds to a Cramer's V of greater than 0.1 for comparisons where the degrees of freedom (df)=1, 0.07, where df=2, 0.06, where df=3, 0.05 and where df=4 or 5.[17]

### Free-text analysis

Free-text responses provided by patients diagnosed with CUP were analysed to provide insights into their experiences of care. Comments were extracted from the CPES data set as individual text files and loaded into the NVivo qualitative data analysis software package. A coding framework for sorting free-text data from a previous study of responses to the Welsh CPES 2013 (WCPES)[18] was

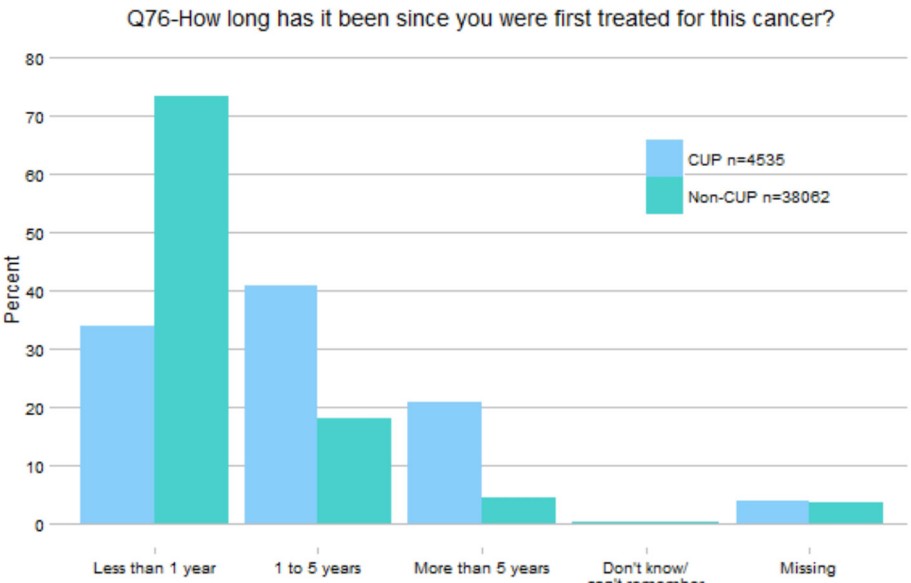

**Figure 1** Percentage of CUP and non-CUP cases for each response category (**Q76**). CUP, cancer of unknown primary.

Table 2 Demographic and clinical characteristics of the sample are shown in table 2

| Observation | Free-text sample (n=3055) | | Marched-pair samples (2992) | | | |
| --- | --- | --- | --- | --- | --- | --- |
| | | | CUP (n=1496) | | Non-CUP (n=1496) | |
| Age | | | | | | |
| Mean (SD) | 65.3 (11.3) | | 66.5 (11.7) | | 66.6 (11.5) | |
| Median (IQR) | 66 (58, 73) | | 67 (59, 75) | | 67 (59, 75) | |
| Range | 16, 95 | | 20, 98 | | 23, 94 | |
| Sex | n | % | n | % | n | % |
| Male | 1121 | 36.7 | 583 | 39.0 | 583 | 39.0 |
| Female | 1934 | 63.3 | 913 | 61.0 | 913 | 61.0 |
| Diagnosis | | | | | | |
| CUP | 3055 | 100 | 1496 | 100 | – | |
| Breast | | | – | | 111 | 7.4 |
| Head and neck | | | – | | 185 | 12.4 |
| Lung | | | – | | 271 | 18.1 |
| Pancreatic | | | – | | 44 | 2.9 |
| Prostate | | | – | | 52 | 3.5 |
| Renal | | | – | | 89 | 5.9 |
| Upper and lower GI | | | – | | 744 | 49.7 |
| ICD-10 codes | | | | | | |
| C77 | 514 | 16.8 | 399 | 27 | – | – |
| C78 | 1172 | 38.4 | 496 | 33 | – | – |
| C79 | 1209 | 39.6 | 435 | 29 | – | – |
| C80 | 160 | 5.2 | 166 | 11 | – | – |

CUP, cancer of unknown primary; ICD-10, International Classification of Diseases 10th Revision.

used. A sample of 200 randomly selected comments were double coded by two researchers (MB, RW) to ensure compatibility of the framework. Coding agreement between the two researchers was 80% (Cohen's Kappa). Any conflicts were resolved through discussion between coders. The framework was then used to categorise the free-text comments. Search criteria were developed for each category, using terms gleaned from term frequency and unique terms analyses of the coded data from the WCPES study. The search strategy was designed to identify relevant comments for each theme. During this process, all comments were read and coded. Following categorisation of comments, a second level of thematic analysis was conducted on specific free-text categories to provide greater depth of insight.[19]

### Ethics and data management

CPES data is a public resource and available from the survey provider, Quality Health. Data were anonymised and secondary analysis of closed questions permitted with agreement from NHS England. Because free-text data may contain personal identifiable data, approval from NHS England to analyse CPES free-text data was separately secured. Free-text analysis of CPES data was approved by the University of Southampton Ethics Committee on 12 November 2014 (UoS Ethics ID:12313).

### FINDINGS

The demographic and clinical characteristics of respondents in both the matched analysis and free-text samples are shown in table 2. The mean age of samples were similar; 65.3 for the CUP free-text sample, 66.5 and 66.6, respectively, for the matched CUP and non-CUP samples. Matched samples contained equal numbers of men (39%, n=583) and women (61%, n=913), proportions were broadly similar for the free-text sample (36.7% (n=1121) and 63.3% (n=1934), respectively). Finally, the ICD-10 codes of respondents indicated c78 (33%, n=496) as the largest categorisation in the CUP matched sample and c79 (39.6%, n=1209) the largest in the free-text sample, while c80 was the smallest categorisation for both samples; 11% (n=166) and 5.2% (n=160), respectively.

### Patient experiences: closed responses

Sixty comparisons were conducted. Of these, nine items showed differences between the CUP and non-CUP respondents that were classified as meaningfully different (table 3). In response to question 8 ('Beforehand, were you given written information about your (diagnostic) tests?'), CUP patients were more likely to respond 'No, but I would have liked written information about the test/s' or 'I did not need written information' and less likely to respond 'Yes, and it was easy to understand'.

**Table 3** Matched paired analysis: CUP and non-CUP patients

| Item | CUP n (%) | Known primary n (%) | Cramer's V |
|---|---|---|---|
| **Q8. Beforehand, were you given written information about your test(s)?** | | | |
| Yes, and it was easy to understand | 808 (83) | 917 (90) | 0.102 |
| Yes, but it was difficult to understand | 37 (4) | 32 (3) | |
| No, but I would have liked written information about the test(s) | 127 (13) | 73 (7) | |
| I did not need written information/Don't know/can't remember | 392 | 334 | |
| **Q9. Were the results of the test(s) explained to you in a way you could understand?** | | | |
| Yes, completely | 961 (72) | 1042 (78) | 0.076 |
| Yes, to some extent | 322 (24) | 259 (20) | |
| No, but I would have liked an explanation | 48 (4) | 28 (2) | |
| I did not need an explanation/Don't know/can't remember/Missing | *33* | *27* | |
| **Q10. Who first told you that you had cancer?** | | | |
| A hospital doctor | 1191 (81) | 1199 (82) | 0.104 |
| A hospital nurse | 44 (3) | 91 (6) | |
| A GP | 155 (11) | 104 (7) | |
| Another health professional | 40 (3) | 51 (4) | |
| A friend or relative | 5 (0.3) | 3 (0.2) | |
| Nobody—I worked it out for myself | 30 (2) | 17 (1) | |
| Missing | 31 | 31 | |
| **Q13. Did you understand the explanation of what was wrong with you?** | | | |
| Yes, I completely understood it | 1006 (68) | 1162 (78) | 0.123 |
| Yes, I understood some of it | 438 (30) | 313 (21) | |
| No, I did not understand it | 32 (2) | 9 (1) | |
| Can't remember/Missing | 20 | 12 | |
| **Q14. When you were told you had cancer, were you given written information about the type of cancer you had?** | | | |
| Yes, and it was easy to understand | 670 (54) | 834 (67) | 0.132 |
| Yes, but it was difficult to understand | 104 (8) | 91 (7) | |
| No, I was not given written information about the type of cancer I had | 466 (38) | 326 (26) | |
| I did not need written information/Don't know/can't remember/Missing | *256* | *245* | |
| **Q16. Do you think your views were taken into account when the team of doctors and nurses caring for you were discussing which treatment you should have?** | | | |
| Yes, definitely | 848 (61) | 951 (68) | 0.078 |
| Yes, to some extent | 343 (25) | 284 (20) | |
| No, my views were not taken into account | 112 (8) | 82 (6) | |
| I didn't know my treatment was being discussed by a team of doctors/nurses | 91 (7) | 80 (6) | |
| Not sure/can't remember/Missing | *102* | *99* | |
| **Q32. During the last 12 months, have you had an operation (such as removal of a tumour or lump) at one of the hospitals in the covering letter?** | | | |
| Yes | 824 (55) | 1004 (67) | 0.124 |
| No | 635 (42) | 469 (31) | |
| Missing | 37 | 23 | |
| **Q34. Beforehand, were you given written information about your operation?** | | | |
| Yes, and it was easy to understand | 472 (64) | 649 (71) | 0.077 |
| Yes, but it was difficult to understand | 36 (5) | 30 (3) | |
| No, I was not given written information about my operation | 229 (31) | 234 (26) | |

**Table 3** Continued

| Item | CUP n (%) | Known primary n (%) | Cramer's V |
|---|---|---|---|
| Don't know/can't remember/Missing | 87 | 91 | |
| Q66. Have you had treatment from any of the following (cancer specialists) for your cancer (patients were asked to tick as many as apply from the following list: physiotherapist; occupational therapist; dietician; speech and language therapist; lymphoedema specialist) | | | |
| Yes | 132 (9) | 37 (3) | 0.138 |

CUP, cancer of unknown primary.

Similarly, Q9 'Were the results of your tests explained to you in a way you could understand?', CUP patients were less likely to respond 'Yes, completely'. In response to Q10 ('Who first told you that you had cancer?'), CUP participants were more likely to respond 'A GP' and less likely to respond 'A hospital nurse', with similar proportions indicating 'a hospital doctor'. In relation to Q13 ('Did you understand the explanation of what was wrong with you?'), CUP patients were less likely to answer 'Yes, I completely understood it' and more likely to respond 'Yes, I understood some of it' or 'No, I did not understand it'. For Q14 (When you were told you had cancer, were you given written information about the type of cancer you had?), CUP patients were less likely to respond 'Yes, and it was easy to understand' and more likely to respond 'No, I was not given written information about the type of cancer I had'. In response to Q16 'Do you think your views were taken into account when the team of doctors and nurses caring for you were discussing which treatment you should have?', CUP patients were less likely to answer 'Yes, definitely'. CUP patients were less likely to report having had surgery in the past 12 months (Q32), and in response to Q34 'Beforehand, were you given written information about your operation?', were less likely to respond 'Yes, and it was easy to understand' and more likely to respond 'No, I was not given written information about my operation'. Finally, CUP patients were more likely to report having received treatment from a lymphoedema specialist (Q66).

### Patient experiences: free text
3055 CUP patients provided comments to one or more of the three free-text questions. The mean length of comments was 64.2 words, with female respondents providing longer comments than male respondents (54.2 words and 69.9 words, respectively). Comments were retrieved from the dataset for 17 categories (table 4). Positive comments were predominant in eight categories ('communication with patients', 'consultants', 'nursing', 'clinical nurse specialists', 'chemotherapy', 'radiotherapy', 'surgery' and 'palliative care'). Predominantly, negative comments were provided by participants for the remaining nine categories ('interagency communication', 'waiting for appointments/investigations', 'waiting on the day', 'receiving results of investigations', 'GPs', 'accident and emergency', 'emotional, social and psychological needs', 'financial concerns' and post-treatment care'). Ratios of negative to positive comments varied widely between categories, with the greatest proportion of negative comments reported for 'waiting time on the day', while the greatest proportion of positive comments was for 'palliative care' (though numbers were small). A trend existed within all themes that positive comments tended to be of a more general quality and scope than negative comments. Essentially, if participants were reporting a negative experience they provided more detail.

Three of the nine closed questions for which responses between CUP and non-CUP patients showed significant differences (Q10, Q13 and Q14) broadly mapped against particular free-text categories. Comments in these categories were subsequently thematically analysed in greater depth.

### Patient information support from health professionals
When investigated further, comments pertaining to –patient information (n=310) provided insights on patient responses to the closed questions 9, 13, 14 and 34; whether patients understood explanations given of their condition or their test results and whether they were provided with sufficient written information about their cancer or tests received. This free-text category had a net ratio of positive comments (0.4:1), but several themes were identified among negative comments (n=89) that indicated why patients found explanations of their condition difficult to understand and why they needed more information. For example, one CUP patient related her difficulty understanding why clinicians were unable to locate the primary tumour site, despite several investigations:

> I had two liver biopsies (the first one did not have sufficient to discern whether benign or malignant). I had a scan while in hospital, also a CT scan and an MRI scan. On my discharge, I later received an appointment for an endoscopy when two small nodules were discovered in the gullet, a biopsy was taken – they were benign, the tumour on the liver is a secondary and they still do not know where the primary is. (Female, 64 years)

Patients were therefore sometimes uncertain or confused with regards the type of cancer they had and

**Table 4** Comment categories with counts and ratios of positive and negative comments

| Comment category | Negative comments (n=) | Positive comments (n=) | Negative to positive ratio (n: 1) | Overall ratio (+ve/−ve) | CUP dataset coverage (%) |
|---|---|---|---|---|---|
| 1. Cross cutting themes | | | | | |
| Interagency communication | 351 | 139 | 2.38 | -ve | 15.3 |
| Patient information | 89 | 221 | 0.40 | +ve | 10.4 |
| Waiting for appointments/investigations | 88 | 72 | 1.24 | -ve | 5.2 |
| Waiting on the day | 299 | 12 | 24.9 | -ve | 10.2 |
| Investigations—receiving results | 165 | 37 | 4.46 | -ve | 6.32 |
| 2. Healthcare professionals | | | | | |
| GPs | 220 | 91 | 2.41 | -ve | 6.32 |
| Consultants | 49 | 98 | 0.50 | +ve | 4.8 |
| Nursing | 284 | 409 | 0.69 | +ve | 22.7 |
| Clinical nurse specialists | 28 | 72 | 0.39 | +ve | 3.3 |
| 3. Treatment specialisms | | | | | |
| Accident and emergency | 28 | 12 | 2.33 | -ve | 1.3 |
| Chemotherapy | 58 | 282 | 0.21 | +ve | 11.1 |
| Radiotherapy | 32 | 81 | 0.39 | +ve | 3.7 |
| Surgery | 170 | 350 | 0.49 | +ve | 17.0 |
| Palliative care | 2 | 40 | 0.05 | +ve | 1.3 |
| Post-treatment care | 38 | 32 | 1.19 | -ve | 2.3 |
| 4. Other quality of life concerns | | | | | |
| Emotional, social and psychological needs | 39 | 17 | 2.29 | -ve | 1.3 |
| Financial concern | 75 | 7 | 10.71 | -ve | 2.7 |

CUP, cancer of unknown primary; GPs, general practitioners.

alluded to the frustration of not knowing and fears of progression:

> Further detailed information of my cancer needed. Exactly where secondaries are and what kind of problems they could cause. Not knowing can be most distressing as you try to second guess things too much. (Female, 72 years)

> They cannot find the primary source so it is rather upsetting as to how the disease is progressing. (Male, 69 years)

Some patients wished to receive more information about what to expect from the diagnosis, but moreover, wanted an opportunity to discuss what this meant for them:

> I would have liked to have had information about how my type of cancer usually progresses, plus a care plan for my own case. I would have liked to have had a chat, as a matter of course, to someone about the practical side of having cancer. (Male, 71 years)

Nevertheless, despite receiving explanations of their conditions from health professionals, patients sometimes reported difficulties with understanding the terminology used:

> The surgeon I see cannot explain things without using clinical terms I do not understand. He could if he tried! And so often I do not understand the detail. In addition he is relatively emotionless. (Female, 57 years)

In addition, patients sometimes reported receiving conflicting information from the many doctors involved in their care:

> Sometimes difficult to get answers as too many doctors and nurses dealing with me and not passing on the relevant information and then having to explain it all over again. (This happened many times.) (Male, 68 years)

Several clinicians could be involved with a CUP patient's care, leading to conflicting information, as they were sometimes passed between clinical teams, as this comment suggests:

> The only thing that could be improved was because it was a process of elimination I went from chest

specialist, to organ oncologist and after my scans and biopsies finally back to my breast oncologist of 11 years ago. So my treatment didn't' start until end of March—6 months later. (Female, 62 years)

## GPs' diagnosis of CUP

Analysis of closed items indicated CUP patients were more likely than non-CUP patients to be given their diagnosis by a GP. Analysis of comments regarding interactions with GPs (n-311) provided by participants found CUP patients were more likely to report negative rather than positive experiences with GPs (ratio of 2.4:1). The theme containing most comments concerning GPs related to speed of diagnosis/referral for further investigations. Negative comments often indicated months and sometimes years of consulting GP services with cancer symptoms before diagnosis and/or specialist referral was made:

> I wish my original GP had listened properly over the months I complained about weight loss (over two stone). Instead I had to change my GP who fast tracked me into hospital where a scan showed metastasized tumours. (Female, 81 years)

Comments often related delayed referrals as a consequence of GPs attributing symptoms to conditions other than cancer or giving insufficient regard to previous cancer diagnoses. Some comments described 'misdiagnosis' of symptoms were later found to be inaccurate.

> Very upset that I have to call the GP, a couple of times a week for 5 weeks, my pain getting worse by the day. My wife asked on a couple of visits if I could have x-ray, blood tests, GP said nothing wrong with me, it was just back ache. In the end my wife took me to A&E. She had to get me into wheelchair from the car. I was in terrible pain. (Male, 79 years)

Poor communication between GP and secondary care services was also a prevalent topic within the 'GP' category and revealed a lack of continuity between GP and secondary care. It is important to note that these comments were often not critical of GP services but of the information provided to them by secondary services.

> I should like my GP to be kept informed more quickly of my treatment at hospital. At the moment, information does not get to her quickly enough, so if I want to discuss something with her, she does not have up to date information, sometimes 3–4 weeks behind. (Female, 65 years)

## DISCUSSION

To our knowledge, these two investigations represent the first, large-scale studies quantifying the experiences of people with CUP compared with people with metastatic known primary cancers and the first qualitative analysis of free-text comments from participants drawn from a national sample of CUP patients. Experiences of patients with CUP were broadly similar to those with advanced metastatic cancer, with the exception of the nine areas reported. This might suggest the success of recommendations advocated by NICE in the UK that relate to the introduction of MDTs and specialist nurses dedicated to patients with CUP,[5] and that equivalent standards of care now exist irrespective of whether the primary site was known or unknown. However, it is not yet clear how consistently CUP MDTs have been established across the UK, and it is doubtful they were well established by 2013.

Patients in the CUP sample were significantly more likely to answer they would have liked more information than the non-CUP sample about the type of cancer they had and the investigations they underwent and significantly less likely to understand the explanations of what was wrong with them. Providing accurate and helpful information and preparing CUP patients for their treatment trajectory is especially difficult given the uncertainty that pervades this diagnosis. The location of the primary tumour is the main reference point for treatment decisions and prognostic information,[20] treatment regimens may change several times during a patient's illness trajectory and patients often receive conflicting information from clinicians.[7] Doctors face challenges when communicating with CUP patients in the face of such uncertainty,[20 21] and consequently doctors often experience discomfort.[22] CUP patients also frequently undergo a greater number of investigations than patients with a known primary, as clinicians sometimes inappropriately 'chase the primary'.[23] Many doctors prefer to estimate a primary site on the basis of clinical signs but there is little consistency in the language used to describe the diagnosis to patients.[21] The number and diversity of health professionals involved with their care is often greater for CUP than non-CUP patients as they are often reviewed and moved between multidisciplinary clinical teams.[7] This can result in confusion and anxiety for patients,[6 7] and may partly explain why CUP patients reportedly felt their views were less often taken into account during treatment decision-making.

CUP patients were significantly more likely to be first told by their GP that they had cancer, compared with the non-CUP sample. It may be that, as CUP patients are often not adopted by a specific MDT or are 'bounced' between MDTs,[7 24] GPs may take responsibility for relaying the diagnosis to the patient on the basis of accumulating evidence, possibly prior to referring them to specialist consultation. Nevertheless, free-text responses also found CUP patients were more likely to report negative rather than positive experiences of interactions with GPs (ratio of 2.4:1), which often related to referrals being delayed as a consequence of GPs attributing symptoms to conditions other than cancer or not sufficiently investigating patients' health concerns. That referral systems for MUO from primary to secondary care is variable across the UK may also contribute to such delays. CUP patients were

significantly less likely to have surgery; by definition, patients with CUP are diagnosed after a primary has metastasised and hence surgery is not usually an option. In contrast, those patients with an advanced known primary may well have had surgery prior to the discovery of metastases. Finally, the finding that patients with CUP are more likely to have seen a lymphoedema specialist is difficult to explain and may be an artefact of the limitations of the sample as discussed below.

A major limitation of this study is that the CUP sample is not representative of the profile of patients with CUP. During 2006–2010, the 1-year relative survival for CUP patients in the UK was 16%,[25] yet in this sample 62% of respondents with CUP had begun treatment for their cancer more than 1 year previous to their diagnosis. Potentially, many of the patients in the CUP sample may have a favourable subtype of CUP: those with likely primary sites that have greater survival rates and reliable treatments,[4] who may be still receiving follow-up care at hospital in the years that followed their original diagnosis. We attempted to control for this potential bias by restricting the CUP sample to those who were first treated for their cancer less than 1 year previously. Nevertheless, these people were still well enough to complete the CPES several months postdischarge. Speculatively, the CUP sample in this analysis may be more like patients with advanced known cancer than a CUP sample recruited prospectively after diagnosis of confirmed CUP, which would have included many more individuals with poorer prognoses. Indeed, while almost half (49%) of patients diagnosed with CUP in 2009 were coded to the ICD-10 code C80, which also accounted for 93% of deaths from CUP in 2010,[26] a much smaller proportion of patients coded C80 were included in the CPES CUP population sample. Therefore, a prospectively recruited sample of CUP patients might have less positive experiences than the CUP sample in this analysis. Moreover, as we selected patients with a known primary for the non-CUP sample to approximate proportions of likely site of origin in CUP cases from autopsy and biomarker studies,[16] we have enriched the non-CUP sample with patients who have poorer prognoses, such as those with pancreatic and lung cancers. These patients may have less positive experiences than a more omnibus sample of patients with advanced known primaries.

Compounding these issues, the diagnoses used in this study were not self-reported or clinician reported but based on administrative data. The ICD-10 classification system does not differentiate between MUO, pCUP and cCUP.[5] For CPES, the ICD code was extracted directly from each Trust's administrative data records, and two errors may subsequently occur with this method. First, as ICD codes are generated by MDT administrators and not by doctors, there may be errors in coding. The second, more serious issue is that because no distinction is made in the data capture between MUO, pCUP and cCUP, it is possible that if a primary site is later confirmed the coding will not be updated in the administrative file. As the survey maybe completed by patients some months postdiagnosis,

some patients within our CUP sample may have already received a site-specific diagnosis, and the sample is therefore not fully representative of the profile of patients with MUO/CUP. Errors in classifying CUP are likely to be more pronounced for the CUP sample due to the uncertainty of whether or when a primary might be found and because patients with MUO can pass among several multidisciplinary teams managing different cancer types.[7] Finally, using administrative data to define the CUP sample also fails to account for the patient's perception of their disease. Indeed, the limited research that exists indicates that patients are often very confused about a diagnosis of CUP and for many patients clinicians convey a 'best guess' primary site to direct treatment or access subsidised medical therapy.[21] What and how they are told of their diagnosis may influence their questionnaire responses.

## CONCLUSION

These findings, when combined, suggest CUP patients may experience delays in diagnosis and to specialist referral and greater uncertainty with regards understanding their diagnosis and be less prepared for what to expect regarding diagnostic investigations than patients with a known primary site. These represent specific areas where targeted psychoeducational interventions might be developed. But, given we identified significant limitations with the CPES CUP sample data, possibly as a consequence of inconsistent or erroneous coding and neglect of the NICE taxonomy of CUP, it is crucial to conduct prospective, observational cohort studies to develop a more complete understanding of the issues faced by CUP patients.

**Author affiliations**
[1]Faculty of Health Sciences, University of Southampton, Southampton, UK
[2]Department of Human Sciences and Public Health, Faculty of Health and Social Sciences, Bournemouth University, Bournemouth, UK
[3]Department of Cancer Experiences Research, Peter MacCallum Cancer Institute, Melbourne, Victoria, Australia
[4]Academy of Research, University of Southampton and University Hospital Southampton NHS Foundation Trust, Southampton, Hampshire, UK
[5]Cancer of Unknown Primary CUP Foundation Jo's Friends, Newbury, West Berkshire, UK
[6]Division of Cancer Medicine, Peter MacCallum Cancer Institute, Melbourne, Victoria, Australia
[7]Department of Psychology, Swinburne University of Technology, Melbourne, Australia

**Acknowledgements** We would like to thank Dr Karla Gough, Senior Research Fellow, Department of Cancer Experiences Research, Peter MacCallum Cancer Centre, who assisted with analysis.

**Contributors** PS, RW and AR conceived the idea of the study and acquisition of the data. PS was responsible for the design of the quantitative study, RW for design of the qualitative study. AD was responsible for undertaking quantitative data analysis. MB and RW were responsible for undertaking analysis of qualitative data. PS, LM, AR, JS, AD, MB and RW were all involved with interpretation of the results from both studies. The initial draft of the paper was produced by RW and MB and circulated repeatedly between all authors for critical revision. All authors read and approved the final manuscript.

**Funding** This work was supported by the Cancer of Unknown Primary (CUP) Foundation and Cancer Australia. Additional funding was provided by the 'Adventures in Research' funding stream at the University of Southampton.

**Competing interests** None declared.

**Provenance and peer review** Not commissioned; externally peer reviewed.

**Data sharing statement** Source data for the study are closed question responses and free text responses to the Cancer Patient Experience Survey in England for 2013. These data are available from the survey provider (Quality Health, UK) https://www.quality-health.co.uk/—email: info@quality-health.co.uk.

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
