## [Reviewer comments · BMJ Open]

ARTICLE DETAILS

TITLE (PROVISIONAL)	Differences in experiences of care between patients diagnosed with metastatic cancer of known and unknown primaries: mixed method findings from the 2013 Cancer Patient Experience Survey in England
AUTHORS	Wagland, Richard; Bracher, Michael; Drosdowsky, Allison; Richardson, Alison; Symons, John; Mileshekin, Linda; Schofield, Penelope

VERSION 1 – REVIEW

REVIEWER	Dr S. Michael Crawford Airedale NHS Foundation Trust United Kingdom
REVIEW RETURNED	26-Jun-2017

GENERAL COMMENTS	This is an interesting topic and the free text results contain some important commentary on health services. The concept of CUP is complex as the authors rightly point out. I suspect that, given the relative longevity of the CUP cohort this study is looking at the C77-C79 ICD grouping and mostly CUP as defined in Table 1 rather than the C80 group which has more sick patients who never get beyond the MUO classification. The authors should confirm this by comparison with cancer registry data. The Phi analysis of Table 3 does not make sense. One figure is quoted for each question whereas the Phi test measures agreement separately for each line; have they calculated a Pearson correlation coefficient? The chi squared statistic for the item in Q10 "first told by a GP" compared to all other listed sources of information is 10.99 (and phi of 0.06) which justifies their statement that CUP patients were significantly more likely to be first told by a GP but this is not evident from the way they present the results and repeated Chi squared testing is not a good way to compare many individual questions. A better way would be to show the proportions for each answer graphically; the 95% confidence intervals for the GP as source answer do not overlap. It is likely to be that the CUP patient is recognised as very likely having metastatic cancer by the GP, for example having a large liver, before referral whereas other patients are referred for investigation of organ-related symptoms and so a specialist practitioner has the duty.
---

	The free-text comments on GPs are important in a healthcare system that lacks diagnostic capacity and which relies on GP gatekeeping to control use of all specialist services. This does not mean that GPs are bad doctors; it means that they are asked to do an impossible task. The information in this paper needs to be published to facilitate discussion around this. Furthermore it would be helpful to see how many individuals and which specialisms are involved in extensive investigation; "chase the primary" is not necessarily good medicine.
--	--

REVIEWER	Ellen Sonet CancerCare United States
REVIEW RETURNED	30-Jun-2017

GENERAL COMMENTS	Note typos and/or words missing: p.2, l. 46 p.5 l.12 p.7 l. 13,21 p.8 l. 24: Add "poor" to beginning of sentence and replace "theme" with "topic" p. 9 l. 29 spelling "artifact" p.10 l.11 replace "between" with "among", l 22 add "to" Comments and Suggestions  1. Why is it relevant that after completing the survey, CUP patients may have had their diagnosis changed to a specific "known" cancer? The important factors for this paper are the experiences and state of mind of the patient who doesn't know what kind of cancer they have. 2. It's clear that when patients have more certainty about their cancer diagnosis and treatment plan, they experience better communication and engagement with their clinical team than when they have CUP. Therefore, even though more research may be needed, it's safe to assume that patients with CUP should be provided with extra psychosocial support to help them cope. 3. The free text questions are very general, regarding "care" from NHS. Patients had very little guidance or structure in answering them, nor is it clear where they were in their treatment journeys at the time they completed the survey. Verbatim comments should be framed as examples of how a specific CUP patient experienced their care without inferring general application to the CUP population.
--

VERSION 1 – AUTHOR RESPONSE

Point No.	Reviewer's comments	Authors' response
	Reviewer 1	
1	The concept of CUP is complex as the authors rightly point out. I suspect that, given the relative longevity of the CUP cohort this study is looking at the C77-C79 ICD grouping and mostly CUP as defined in Table 1 rather than the C80 group which has more sick patients who never get beyond the MUO classification. The authors should confirm this by comparison with cancer registry data.	As we note (para 4, p9), the ICD-10 classification system does not differentiate MUO, pCUP or cCUP, and different Trusts code for CUP in different and inconsistent ways. However, the reviewer is correct to suggest that fewer participants were coded as c80 in our samples (5.2% of free text responders and 11% of those included within the matched analysis) than c77-79. These figures have been added to table 2 giving participant's characteristics, and a paragraph added at the beginning of the Findings section to highlight the demographic and clinical characteristics of respondents. This issue with ICD proportions is later picked up in the Discussion, with an added sentence comparing the ICD codes of respondents with the National Cancer Intelligence Network (NCIN) figures. 'Indeed, while almost half (49%) of patients diagnosed with CUP in 2009 were coded to the ICD-10 code c80, which also accounted for 93% of deaths from CUP in 2010 [16], a much smaller proportion of patients coded c80 were included in the CPES CUP population sample.'
2	The Phi analysis of Table 3 does not make sense. One figure is quoted for each question whereas the Phi test measures agreement separately for each line; have they calculated a Pearson correlation coefficient?	Thanks for this comment. The phi coefficient provides a measure of association for 2-by-2 tables; Cramer's V is used with everything else. The estimates themselves were actually identical, but cut-offs for small-, medium- and large-sized associations are slightly different (Cohen J. Statistical power analysis for the behavioural sciences. New York, NY: Lawrence Erlbaum Associates, 1988). This was an oversight on our behalf, which we have now corrected. We reappraised the phi/Cramer's coefficients for all questions; this turned up three extra items where the association could be characterised as small-sized. These include: Q9 'Were the results of the test(s) explained to you in a way you could understand?' More CUP patients said no (72% vs 78% non-CUP). Q16 'Do you think your views were taken into account when the team of doctors and nurses caring for you were discussing which treatment you should have?' Fewer CUP patients said "Yes, definitely", and correspondingly more said "Yes to an extent" or "No". Q34 'Beforehand, were you given written information about your operation?' More CUP patients reported they were not given written information. We have added these items to Table 3, and revised

		the text in the 'Patient experiences: closed responses' section (p6). These additional items are similar to three items already reported in the first draft of this paper, i.e. those relating to CUP patients experiencing less written information regarding the 'tests' (Q8) and their type of cancer (Q13), and explanations regarding their condition (Q14). The existing Conclusion does not therefore need to be revised. Reference to these additional items has been added to the Results section where the free-text is described (under 'Patient information support from health professionals' sub-heading) (p7): 'When investigated further, comments pertaining to staff-patient communication (n=310) provided insights on patient responses to the closed questions Qs 9, 13, 14 and 34; whether patients understood explanations given of their condition or their test results, and; whether they were provided with sufficient written information about their cancer or tests received.' The finding that CUP patients are less likely than non-CUP patients to report their views were 'taken into account' during treatment decision-making may be related to inconsistencies of language often used to describe a CUP diagnosis, the greater number and diversity of health professionals involved with their care, and the sometimes frequent moves between multiple MDTs. The text (para 1, p9) has been slightly revised to argue this: 'This can result in confusion and anxiety for patients [6,7], and may partly explain why CUP patients reportedly felt their views were less often taken into account during treatment decision making.'
3	The chi squared statistic for the item in Q10 "first told by a GP" compared to all other listed sources of information is 10.99 (and phi of 0.06) which justifies their statement that CUP patients were significantly more	The chi-square test is a standard method for examining the association between two categorical variables (Agresti A. Categorical data analysis 2nd ed. Hoboken, NJ: John Wiley & Sons), but the results of the significance test tell us little about the nature of the association. We have therefore followed up with

	likely to be first told by a GP but this is not evident from the way they present the results and repeated Chi squared testing is not a good way to compare many individual questions. A better way would be to show the proportions for each answer graphically; the 95% confidence intervals for the GP as source answer do not overlap.	the phi-coefficient; to characterise the size of the association (and whether it was likely to have any substantive significance). We also examined the standardised Pearson residuals to identify cells having lack of fit (larger or smaller than expected frequencies), and used these to inform how these associations were described in-text in the 'Frequency matching analysis' section (p5). The residuals themselves were not presented, as they are not easily interpreted and require a lot of explanation.
4	It is likely to be that the CUP patient is recognised as very likely having metastatic cancer by the GP, for example having a large liver, before referral whereas other patients are referred for investigation of organ-related symptoms and so a specialist practitioner has the duty.	Thank you. We agree that patients may be warned of possible mets in this way, and patients may remember this when responding to the CPES. A sentence (underlined) has been added to the text (p10) to indicate this: ' GPs may take responsibility for relaying the diagnosis to the patient, on the basis of accumulating evidence. GPs may also warn patients of a likely metastatic cancer, possibly prior to referring them to specialist consultation. '
5	The free-text comments on GPs are important in a healthcare system that lacks diagnostic capacity and which relies on GP gatekeeping to control use of all specialist services. This does not mean that GPs are bad doctors; it means that they are asked to do an impossible task. The information in this paper needs to be published to facilitate discussion around this.	Thank you. We recognise that GPs are not bad doctors but have very high workloads. We also realise that variable referral systems across UK networks may contribute to delays, and have added the following sentence in the Discussion where we reflect on negative patient comments about referral delays: 'That referral systems for MUO from primary to secondary care is variable across the UK may also contribute to such delays.' We are aware that a 2WW referral working group is presently studying this issue, specifically in relation to MUO/CUP, as part of The Cancer Strategy Implementation Support- Early Diagnosis team (part of Transforming Cancer Services Team for London).
6	Furthermore it would be helpful to see how many individuals and which specialisms are involved in extensive investigation; "chase the primary" is not necessarily good medicine.	We are unable to report the number of individuals and the specialisms that had most extensive investigations in this paper, as the free text data were anonymised and not linked to secondary tumour sites. Moreover, over-investigation is identified via the free-text data, but individuals may have been over-investigated and do not commented on this, meaning

		it would be difficult to quantify this meaningfully. We agree that to chase the primary is not necessarily 'good medicine', and have clarified this in the text (1st paragraph, p9) to read 'clinicians sometimes inappropriately 'chase the primary''
	Reviewer 2	
7	Note typos and/or words missing:  > p.2, l. 46 > p.5 l.12 > p.7 l. 13,21 > p.8 l. 24: Add "poor" to beginning of sentence and replace "theme" with "topic" > p. 9 l. 29 spelling "artifact" > p.10 l.11 replace "between" with "among", l 22 add "to" 	Thank you. These corrections have been made.
8	Why is it relevant that after completing the survey, CUP patients may have had their diagnosis changed to a specific "known" cancer? The important factors for this paper are the experiences and state of mind of the patient who doesn't know what kind of cancer they have.	The following sentence has been added to the last para of the Discussion (p10) to explain the relevance of patients being given a primary site diagnosis prior to completing the survey: 'As the survey maybe completed by patients some months post-diagnosis, some patients within our CUP sample may have already received a site-specific diagnosis, and the sample is therefore not fully representative of the profile of patients with MUO/CUP.'
9	It's clear that when patients have more certainty about their cancer diagnosis and treatment plan, they experience better communication and engagement with their clinical team than when they have CUP. Therefore, even though more research may be needed, it's safe to	We agree with this observation from the reviewer. As we indicate in the Introduction (paras 2-3, p2), the unique psycho-social needs of MUO/CUP patients have started to be recognised by researchers. We do not think the text needs revising in the paper, but please let us know if you disagree.

	assume that patients with CUP should be provided with extra psychosocial support to help them cope.	
10	The free text questions are very general, regarding "care" from NHS. Patients had very little guidance or structure in answering them, nor is it clear where they were in their treatment journeys at the time they completed the survey. Verbatim comments should be framed as examples of how a specific CUP patient experienced their care without inferring general application to the CUP population.	The free-text questions are indeed very general, and themes were developed from the data when numbers of patients provided comments that said similar things. Because there is little guidance for patients as to what they should write about, the fact that many chose to write comments about similar experiences justifies describing them as 'themes'. However, we have revised the text in two places to indicate comments were not universal when previously the text may have indicated they were: 'For example, one CUP patients related her difficulty understanding why clinicians were unable to locate the primary tumour site, despite several investigations.' (p7) 'Some Comments described 'misdiagnosis' of symptoms later found to be inaccurate.' We believe the text that describes the other comments indicates that only 'some' patients or patients 'sometimes' responded by describing such experiences.

VERSION 2 – REVIEW

REVIEWER	Dr S. Michael Crawford Airedale NHS Foundation Trust, BD20 6TD, UK
REVIEW RETURNED	08-Aug-2017

GENERAL COMMENTS	The previous points raised in review have been addressed.
---

REVIEWER	Ellen Sonet CancerCare, United States
REVIEW RETURNED	07-Aug-2017

GENERAL COMMENTS	Thank you for addressing concerns of this reviewer. Nice paper.
---